# Computational method for aromatase-related proteins using machine learning approach

**Muthu Krishnan Selvaraj[1]\*, Jasmeet Kaur[2]\***

**1** Data Center/Bioinformatics, MTCC, CSIR-Institute of Microbial Technology, Chandigarh, India,
**2** Department of Biophysics, Postgraduate Institute of Medical Education and Research (PGIMER), Chandigarh, India

\* muthu@imtech.res.in (MKS); jasmeet23k@gmail.com (JK)

## Abstract

Human aromatase enzyme is a microsomal cytochrome P450 and catalyzes aromatization of androgens into estrogens during steroidogenesis. For breast cancer therapy, third-generation aromatase inhibitors (AIs) have proven to be effective; however patients acquire resistance to current AIs. Thus there is a need to predict aromatase-related proteins to develop efficacious AIs. A machine learning method was established to identify aromatase-related proteins using a five-fold cross validation technique. In this study, different SVM approach-based models were built using the following approaches like amino acid, dipeptide composition, hybrid and evolutionary profiles in the form of position-specific scoring matrix (PSSM); with maximum accuracy of 87.42%, 84.05%, 85.12%, and 92.02% respectively. Based on the primary sequence, the developed method is highly accurate to predict the aromatase-related proteins. Prediction scores graphs were developed using the known dataset to check the performance of the method. Based on the approach described above, a webserver for predicting aromatase-related proteins from primary sequence data was developed and implemented at https://bioinfo.imtech.res.in/servers/muthu/aromatase/home.html. We hope that the developed method will be useful for aromatase protein related research.

## Introduction

Cancer cases continue to rise globally despite advances in clinical therapy [1]. Breast cancer remains the most frequently diagnosed cancer in females and metastasis remains the leading cause of death by this cancer [1]. Breast cancer incidence is greater in developed countries, while mortality is highest in developing countries [2]. About 30% breast cancer patients develop recurring metastatic cancer despite recent advances in therapeutic regimens.

Biological actions of estrogen are mediated with the estrogen receptor (ER) and 70% of breast tumors express the ER and/or progesterone receptor (PR). Thus, estrogen deprivation has been considered an important treatment for estrogen-dependent (ER+) breast cancers. In post-menopausal women, estradiol is produced in extragonadal sites and thus it stops functioning as a circulating hormone and acts locally as a paracrine or intracrine factor [3, 4]. These peripheral sites include the mesenchymal cells of adipose tissue, osteoblasts

**Data Availability Statement:** All relevant data are within the paper and it Supporting information files.

**Funding:** The authors received no specific funding for this work.

**Competing interests:** The authors have declared that no competing interests exist.

and chondrocytes of bone and numerous sites in the brain and promotes breast cancer [5, 6].

For a long time, tamoxifen has been a reliable therapeutic measure for ER+ breast cancer, in both pre- and post-menopausal women. However, over half of advanced ER+ breast cancers are intrinsically resistant to tamoxifen and about 40% will acquire the resistance during the treatment. Aromatase inhibitors (AIs) are the next line of therapeutic approach for ER+ breast cancer in women and serve as first-line therapy for metastatic breast cancer [7]. AIs block the action of microsomal aromatase cytochrome P450 (P450arom), thus limiting estrogen biosynthesis and tumor progression [8]. Aromatase is a product of *CYP19A1* gene, which produces a monomeric enzyme composed of a heme group and a single polypeptide chain of 503 amino-acids [9, 10]. Aromatase is primarily expressed in gonads and brain of humans, but also occurs in placenta and liver of developing fetus, and in muscle, adrenal cortex and adipose tissue of the adults [9]. In the ovary, aromatase is produced in the granulosa cells and converts androgens (male hormones) into estrogens (female hormones) and is essential for the female reproductive cycle, development of female secondary sexual characteristics and for maintaining reproductive health [11].

AIs are currently an established treatment regimen for the ER+ breast cancer patients and FDA has approved first-, second-, and third-generation AIs. The third-generation inhibitors including letrozole, anastrozole and exemestanea are routine treatment for post-menopausal breast cancer patients [12]. Besides the therapeutic success of the third-generation AIs, acquired resistance develops, leading to tumor relapse [13]. Further, patients with prolonged clinical usage of both steroidal and non-steroidal third-generation AIs, have experienced side effects like myalgia, arthralgia, hot flashes and night sweats [14]. Thus, there is an urgency to develop novel aromatase inhibitors for improved effectiveness and lesser side-effects.

Machine learning is coming-up as a useful tool in biological science [15] and it can be used to uncover novel aromatase-related proteins and to investigate the structural and functional properties of these enzymes. At present many computational methods has been proposed for clinical data analysis, clinically important protein or enzymes using machine learning approaches [16–19]. But so far, there are no reports on investigating aromatase-related proteins by support vector machine (SVM). Thus finding or identifying novel or unknown aromatase-related proteins using SVM, is the need of the hour.

Support vector machine is a governed machine learning method, commonly used in bioinformatics applications, such as predicting protein functions and their evolutionary correlations, analyzing DNA sequences, and classifying microarray data [20–22]. Due to its powerful prediction ability, it is used not just for protein studies but in numerous clinical investigations like gene expression profiling, cancer classification and biomarker discovery [23]. SVM is also being employed for prediction of drug-target interactions, disease-associated genes and drug efficacy [24]. Its use in gene selection and classification of microRNA expression data has enabled researchers to analyze large datasets and help understand relationship between genes and diseases [25].

SVM statistical predictors for sequence-based biological system use step-wise rules: dataset construction/selection to coach and examine a predictor, programming the biological sequence in an effective mathematical term, developing a vigorous algorithm to run the prediction, performing cross-validation to evaluate prediction accuracy and running the algorithm on a public accessible user-friendly web-server [26]. It is one of the major tasks in bioinformatics to predict the protein functions using protein structure, post-translational modification (PTM) sites and DNA binding sites; which can assist in understanding disease mechanisms and/or identifying novel drug targets [23, 27, 28].

Therefore, we have made a concerted effort to develop a method for identifying aromatase-related proteins. We developed a method for recognizing enzymes that will aid in the identification of new or unknown aromatase-related proteins, using amino acid composition (AAC), dipeptide composition (DPC), hybrid and position-specific scoring matrix (PSSM) models.

## Methods

### Machine learning based support vector machine (SVM)

Amino acid composition (AAC), dipeptide composition (DPC), PSSM profile and Hybrid approach employing machine learning based support vector Machine (SVM) were used to construct the method. The SVM-based prediction technique is often used to manage vast amounts of data, and it has been demonstrated to perform well in a number of biological data processing applications such as classification, protein functions and type identification [29–31]. In this study, we used SVM to analyze the performance of the classifiers and five-fold cross validation [32–34]. The generated approach model's performance was assessed using the original and additional protein datasets. To eliminate outcome bias, all models were run with the same amount of negative sequences. Based on the size of the aromatase dataset, negative sequences were picked at random from the UniProt database. The performance of the SVM models was tested using known positive and negative sequence data. A blank dataset was also utilized to test the generated models, which successfully recognized the data.

### Generation of survival curves

Kaplan-Meier (KM) plotter is a web-based survival analysis tool and evaluates correlation between the expression of all genes (mRNA, miRNA, protein) and survival in about 30k+ samples from all tumor types. GEO, EGA, and TCGA are the sources for the databases and the plotter provides a meta-analysis based discovery and validation of survival biomarkers for cancer research [35]. The KM plotter tool (http://kmplot.com/analysis/) was used to determine the prognostic value of aromatase *(CYP19A1)* mRNA expression using Pan-cancer RNA-seq in various cancers by correlating it with overall (OS) and relapse-free (RFS) survival [36], for a follow-up threshold of 240 months. For mRNA expression analysis, samples were split into high and low expression groups based on the median expression of aromatase. The median expression was selected to split patients over other options of lower quartile, lower tertile, upper tertile and upper quartile expression to give almost same sample numbers for both groups and hence less bias. Hazard ratio (HR), 95% confidence intervals and logrank p for all the survival curves were provided by the KM plotter website and p value of $< 0.05$ was considered to be statistically significant.

### Datasets for SVM

Aromatase data was taken from the Uniprot/SWISSPROT database [37]. When we used the keyword, we found 9836 protein sequences which included 257 reviewed sequences. So, we used only reviewed sequences retrieved on 10th May 2021, and removed all these sequences annotated or labeled as "fragments," "isoforms," "potentials," "similarity," or "probables" to generate a high quality dataset and this removal will help in reducing the prediction error. To avoid redundancy and the incorporation of variants, this dataset was then processed with the CD-hit tool, which deleted sequences that were more than 90% identical to any other sequence in the dataset [38]. The final dataset contained a total of 191 aromatase sequences (positive dataset) out of 257, details provided in the S1 File. The negative dataset contained 191 non-aromatase sequences that were unrelated to the aromatase and were picked at random. A

Uniprot/Swissprot keyword search for "regulatory proteins" was used to select the negative sequence collection. A web server for predicting aromatase-related proteins from primary sequence data was developed and implemented at weblink https://bioinfo.imtech.res.in/servers/muthu/aromatase/home.html.

## Amino acid and dipeptide composition

The amino acid composition of a protein refers to the percentage of each amino acid in the protein [21, 39]. Encoding data into vectors is required by the SVM light. The percentage of all 20 natural amino acids was calculated using the following equation:

$$\text{Fraction of amino acid (i)} = \frac{\text{Total number of amino acid (i)}}{\text{Total number of amino acid in protein}} \quad (1)$$

In a similar manner, dipeptide composition was calculated using a vector with a constant length of 400 (20x20) dimensions [40]. To determine the fraction of each dipeptide composition, the following equation was used:

$$\text{Fraction of dipeptide (i)} = \frac{\text{Total number of dipep (i)}}{\text{Total number of all possible dipeptides}} \quad (2)$$

## PSSM profile

The GPSR software was used to create the PSSM profile against the nr (non-redundant) blast database. We utilized the seq2pssm imp, pssm n2, pssm comp, and col2svm programmes in the GPSR package for PSI-BLAST searches against the nr database using different iterations with a cut-off e-value of 0.001, as well as to normalize the PSSM profile and produce the SVM light input format (i.e. as a composition vector of 400) [26]. Finally, the SVM models were created with various parameters, optimized, and the best model was employed in the prediction server. For normalization, the following formula was used:

$$\text{Normalized value} = \frac{(\text{Value} - \text{Minimum})}{(\text{Maximum} - \text{Minimum})} \quad (3)$$

## Hybrid approach

In order to improve prediction accuracy, a hybrid technique was developed. A hybrid model is defined as the combination of two or more profiles. The hybrid models were developed using 420 vector lengths, which included 20 and 400 from AAC and DPC, respectively. The *col_add* function in the GPSR 1.0 package's was used to merge the AAC and DPC profiles to generate a hybrid profile [41, 42].

## Evaluation and performance

A five-fold cross validation approach was used to evaluate performance. We started with an aromatase positive dataset and a non-aromatase negative dataset. Positive and negative datasets were randomly divided into five equal groups. In order to run SVM, four sets were utilized for training and the remaining set for testing. This process was performed five times, resulting in only one test for each sub-set [22, 43]. This has been done with all approaches, including amino acid, dipeptide, PSSM, and hybrid. The average of the test scores from all five sets was used to compute the final performance. The performance of the classifiers was assessed using sensitivity, specificity, accuracy, and the Mathew correlation coefficient (MCC). These

measurements were calculated using the following standard formulas:

$$\text{Accuracy (ACC)} = \frac{\text{TP} + \text{TN}}{\text{TP} + \text{TN} + \text{FP} + \text{FN}} \qquad (4)$$

$$\text{Sensitivity (SN)} = \frac{\text{TP}}{\text{TP} + \text{FN}} \qquad (5)$$

$$\text{Specificity (SP)} = \frac{\text{TN}}{\text{TN} + \text{FP}} \qquad (6)$$

$$\text{MCC} = \frac{\text{TP X TN} - \text{FP X FN}}{\sqrt{(\text{TP} + \text{FP})(\text{TP} + \text{FN})(\text{TN} + \text{FP})(\text{TN} + \text{FN})}} \qquad (7)$$

### Support vector machine (SVM)

Aromatase prediction was done with the SVM light programme, a very successful machine learning approach. The SVM-light has been used in a variety of investigations, including plasminogen activator prediction, BacHbpred-bacterial hemoglobin prediction, Oxypred-oxygen-binding protein prediction, and VerHb-vertebrate hemoglobin protein prediction [21, 26, 39–42]. The SVM may employ a range of parameter settings, including kernel, linear, polynomial, and radial basic functions (RBI) [44]. We optimized distinct parameters for each prediction approach in the prediction studies. In the method, aromatase was utilized as a positive example and non-aromatase was used as a negative example. In practice, we ran SVM light with (+)ve labels for positive sequences and (-)ve labels for negative sequences.

### Webserver

The aromatase related protein prediction webserver was developed using HTML and CGI--PERL script. The backend was connected to the apache server utilizing the linux operating system. The prediction webserver can be accessed freely at the following weblink https://bioinfo.imtech.res.in/servers/muthu/aromatase/home.html. It is a Support Vector Machine (SVM) based classification method for predicting aromatase-related protein. The user can paste their sequences in fasta format into the text box on the submit page. This server will predict the input sequences as aromatase or non-aromatase protein, based on the selected approaches—amino acid composition (AAC), dipeptide composition (DPC), PSSM and hybrid (AAC+DPC).

## Results

### Effect of aromatase mRNA expression on cancer patient's survival

KM plotter Pan-cancer RNA-seq was used to analyze correlation of aromatase (*CYP19A1*) mRNA expression and survival in different available tumor types (Table 1). Aromatase higher mRNA expression significantly correlated to poorer OS in head-neck squamous cell carcinoma (Fig 1A, Table 1), kidney renal clear cell carcinoma (Fig 1B, Table 1) and kidney renal papillary cell carcinoma (Fig 1C, Table 1) patients. Aromatase higher mRNA expression was also significantly correlated to poorer RFS in kidney renal papillary cell carcinoma (Fig 1D, Table 1) patients. Further, higher aromatase mRNA expression led to significantly poorer OS in liver hepatocellular carcinoma (Fig 1E, Table 1) and stomach adenocarcinoma (Fig 1F,

**Table 1. Correlation of aromatase mRNA expression with overall (OS) and relapse-free survival (RFS) in various cancer patients.**

| Tumor type | Samples with RNAseq data | OS | | | RFS | | |
|---|---|---|---|---|---|---|---|
| | | HR | 95%CI | p-value | HR | 95%CI | p-value |
| Bladder Carcinoma | 405 | 1.05 | 0.78 – 1.41 | 0.75 | 1.22 | 0.6 – 2.48 | 0.59 |
| Breast cancer | 1090 | 1.08 | 0.78 – 1.48 | 0.66 | 1.34 | 0.87 – 2.06 | 0.19 |
| Cervical squamous cell carcinoma | 304 | 1.33 | 0.83 – 2.12 | 0.24 | 0.51 | 0.23 – 1.15 | 0.1 |
| Esophageal Adenocarcinoma | 80 | 1.4 | 0.74 – 2.66 | 0.3 | 1.09 | 0.15 – 7.93 | 0.93 |
| Esophageal Squamous Cell Carcinoma | 81 | 1.06 | 0.48 – 2.33 | 0.89 | 1.7 | 0.65 – 4.47 | 0.28 |
| Head-neck squamous cell carcinoma | 500 | 1.38 | 1.06 – 1.8 | 0.017 | 2.15 | 0.99 – 4.67 | 0.048 |
| Kidney renal clear cell carcinoma | 530 | 1.48 | 1.1 – 2 | 0.01 | 0.61 | 0.22 – 1.71 | 0.34 |
| Kidney renal papillary cell carcinoma | 288 | 2.11 | 1.13 – 3.94 | 0.017 | 2.3 | 1.07 – 4.96 | 0.028 |
| Liver hepatocellular carcinoma | 371 | 1.77 | 1.25 – 2.5 | 0.001 | 1.1 | 0.79 – 1.53 | 0.56 |
| Lung adenocarcinoma | 513 | 1.27 | 0.95 – 1.7 | 0.11 | 1.3 | 0.86 – 1.97 | 0.22 |
| Lung squamous cell carcinoma | 501 | 1.3 | 0.99 – 1.7 | 0.061 | 1.18 | 0.71 – 1.94 | 0.53 |
| Ovarian cancer | 374 | 0.97 | 0.74 – 1.25) | 0.79 | 0.97 | 0.69 – 1.38 | 0.88 |
| Pancreatic ductal adenocarcinoma | 177 | 0.89 | 0.59 – 1.35 | 0.59 | 1.25 | 0.53 – 2.93 | 0.61 |
| Pheochromocytoma and Paraganglioma | 178 | 3.27 | 0.58 – 18.55 | 0.16 | 0.5 | 0.05 – 4.82 | 0.54 |
| Rectum adenocarcinoma | 165 | 0.93 | 0.43 – 2 | 0.84 | 4.35 | 0.5 – 37.68 | 0.15 |
| Sarcoma | 259 | 1.14 | 0.77 – 1.7 | 0.52 | 1.42 | 0.87 – 2.32 | 0.16 |
| Stomach adenocarcinoma | 375 | 1.41 | 1.02 – 1.95 | 0.038 | 1.43 | 0.75 – 2.74 | 0.28 |
| Testicular Germ Cell Tumor | 134 | 2.07 | 0.19 – 22.89 | 0.54 | 1.1 | 0.52 – 2.33 | 0.81 |
| Thymoma | 119 | 1.16 | 0.3 – 4.41 | 0.83 | Sample number too low for meaningful analysis | | |
| Thyroid carcinoma | 502 | 0.76 | 0.28 – 2.1 | 0.6 | 0.74 | 0.34 – 1.63 | 0.45 |
| Uterine corpus endometrial carcinoma | 543 | 1.07 | 0.71 – 1.61 | 0.75 | 0.9 | 0.54 – 1.52 | 0.71 |

Table 1) patients. No significant correlation between aromatase mRNA expression and survival was seen for other types of tumors (Table 1).

## Amino acid composition analysis

The amino acid composition of aromatase sequences was computed for aromatase proteins, and it was observed that residue "L" occurs at much greater frequencies (above 10%) (Fig 2A). As shown in Fig 2A, "F", "P", "S" and "V" are present more than 6%. The residues "C" and "W" are shown less than 2%. When comparing the amino acid residue profiles of aromatase and non-aromatase, some of the residues pattern are similar, but not all (Fig 2B). These differences can be used to identify the aromatase from negative sequence by the developed models.

## Amino acid composition SVM modules

Firstly we used support vector machines (SVM) to develop models based on the amino acid composition of aromatase. SVM was trained on a variety of datasets using the SVM light implementation. A 20-dimensional amino acid composition vector was used to train the SVM classifiers. SVM Kernels and parameters were adjusted for the best discriminating between positive and negative protein sequence data sets. The maximum accuracy (ACC) of aromatase prediction based on amino acid composition was 87.42%, with 100% sensitivity (SN), 74.84% specificity (SP) and 0.87 Mathew correlation coefficient (MCC) (Table 2, Fig 3).

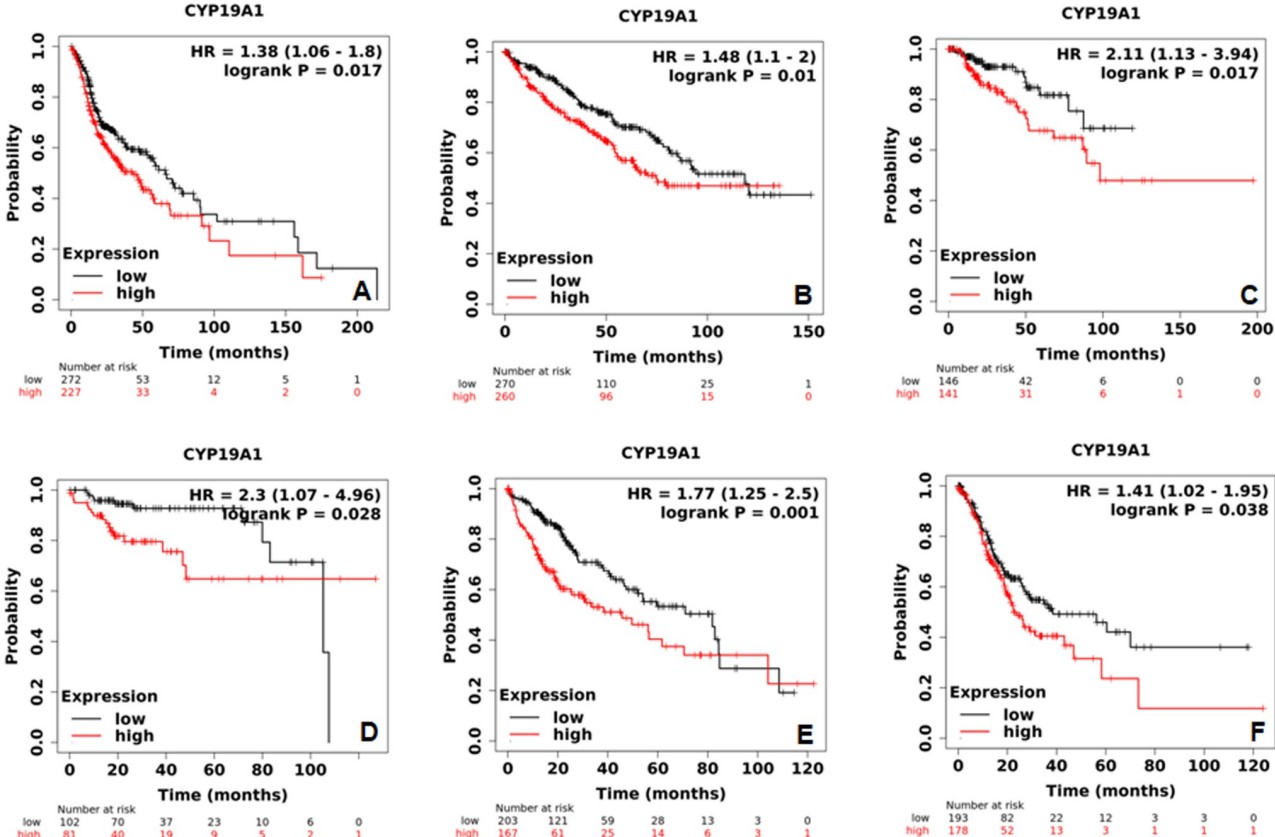

**Fig 1.** Effect of aromatase mRNA expression on OS in head-neck squamous cell carcinoma (A), kidney renal clear cell carcinoma (B) and kidney renal papillary cell carcinoma (C) patients. (D) Effect of aromatase mRNA expression on RFS in kidney renal papillary cell carcinoma. Effect of aromatase mRNA expression on OS in liver hepatocellular carcinoma (E) and stomach adenocarcinoma (F) patients.

## SVM modules using dipeptide composition

In general, SVM algorithms based on dipeptide composition are more effective than approaches based on single amino acid composition. SVM classifiers for dipeptide composition have also been constructed, which is represented by a 400-dimensional vector of dipeptide frequencies (20 x 20). During the adjustment of the kernel parameter and trade-off parameter C, better prediction performance was found with $\gamma = 3$ and $C = 375$. We developed models to distinguish aromatase from non-aromatase sequences based on these parameters. The SVM-based model achieved a maximum accuracy of 84.05%, 99.84% sensitivity, 68.26% specificity and 0.82 MCC as shown in Table 2 and Fig 3.

## Hybrid (AC + DC) SVM modules

The aromatase prediction problem was also addressed using a hybrid prediction approach that integrated amino acid composition (AAC) and dipeptide composition (DPC). The hybrid approach yielded 85.12% accuracy, 98.68% sensitivity, 71.55% specificity and 0.83 MCC respectively (Table 2, Fig 3). The hybrid model results are slightly improved than the individual models, the hybrid model increase sensitivity while decrease in specificity, resulting in a slight improvement in overall performance.

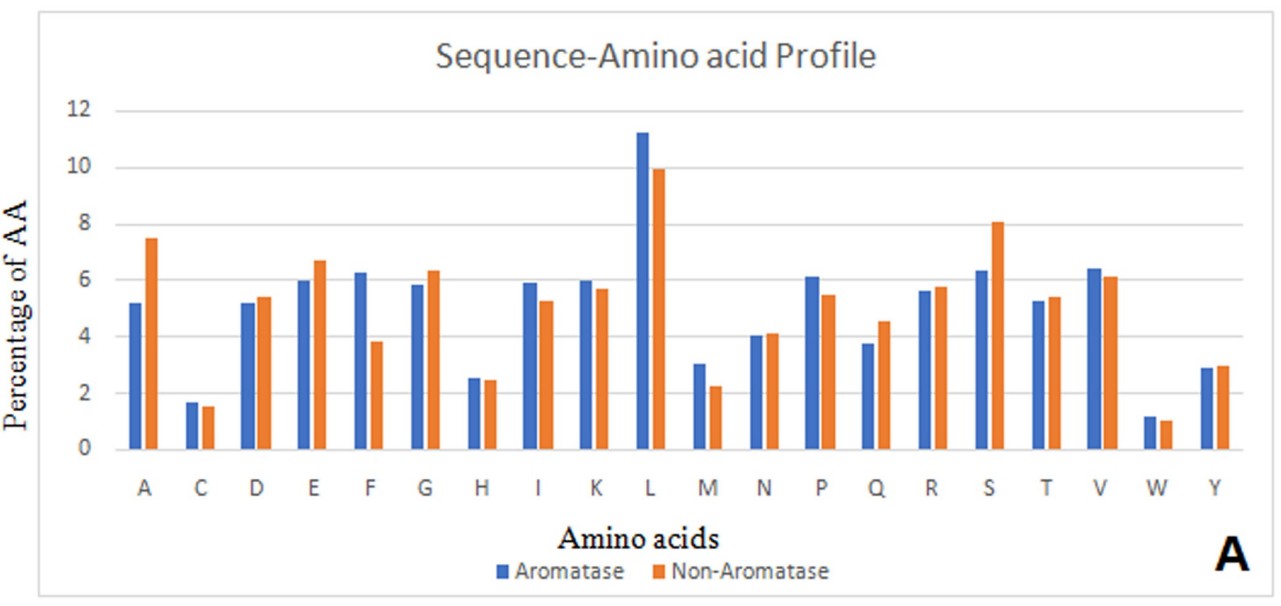

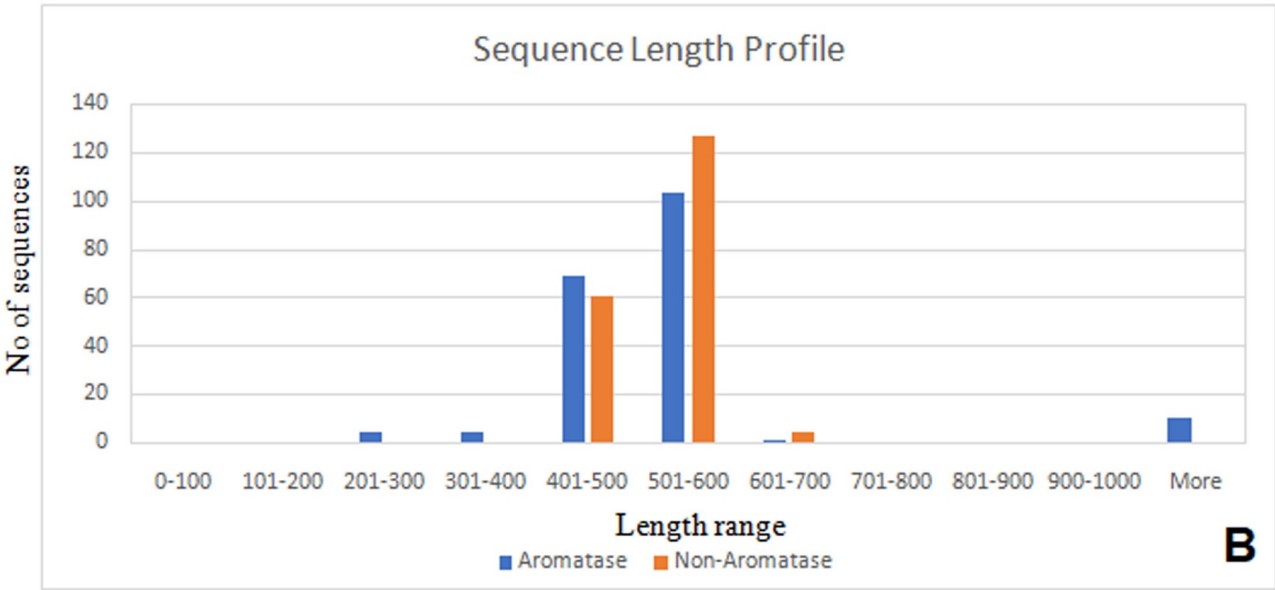

**Fig 2.** A) Amino acid distribution chart between aromatase and non-aromatase protein sequences. B) Sequence length profile of aromatase and non-aromatase proteins binding.

**Table 2. The performance of SVM models using AAC, DPC, Hybrid and PSSM profiles on the original datasets.**

|        | ACC (%) | SN (%) | SP (%) | MCC  |
|--------|---------|--------|--------|------|
| AAC    | 87.42   | 100    | 74.84  | 0.87 |
| DPC    | 84.05   | 99.84  | 68.26  | 0.82 |
| Hybrid | 85.12   | 98.68  | 71.55  | 0.83 |
| PSSM   | 92.02   | 100    | 84.05  | 0.92 |

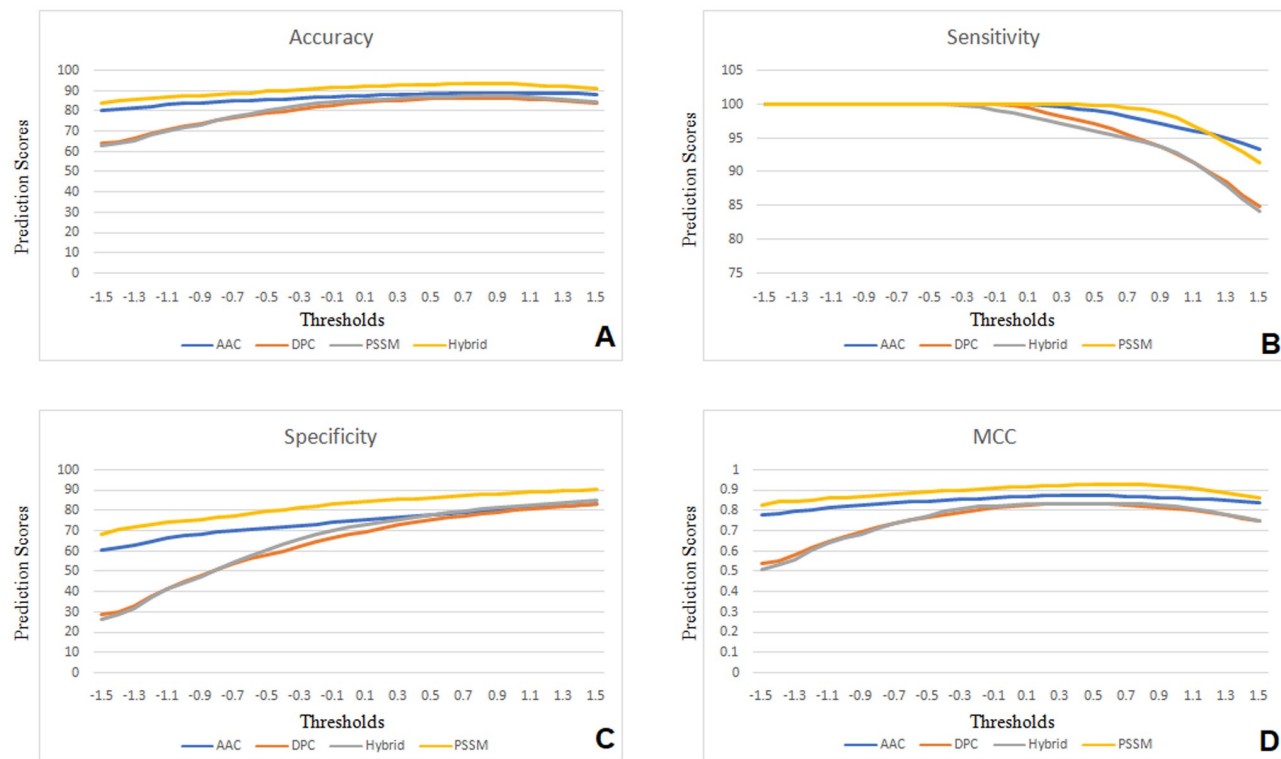

**Fig 3. The performance of accuracy (A), sensitivity (B), specificity (C) and MCC (D) based on the threshold value in all approaches.**

## PSSM profile based SVM modules

Aromatase prediction models based on position specific score matrix (PSSM) profiles were also developed to improve the performance, and they achieved maximum accuracy of 92.02% with 100% sensitivity, 84.05% specificity and 0.92% of MCC (Table 2, Fig 3). In general, all models, including the simple AAC method, performed comparably well as measured by accuracy and MCC.

## Prediction scoring graphs analysis

Prediction scoring graphs were also used to assess the performance of SVM modules. The prediction score for each individual sequence tested is represented by the scoring graph, which shows how the score of sequences in the positive set is separated from the score of sequences in the negative set by a threshold that may be used to categorize positive and negative predictions. However, not all positive or negative sequences are successfully categorized, leading to misleading negative and positive predictions. This analysis summarizes the prediction results to reflect this element of performance. According to our study's findings, no positive sequences predicted negatively in AAC, whereas one negative sequence predicted positively (Fig 4A). In DPC, no positive sequences predicted negatively and no negative sequences predicted positively (Fig 4B). In hybrid, three positive sequences predicted negatively while one negative sequence predicted positively (Fig 4C). One positive sequence predicted negatively whereas the one negative sequence predicted positively in the PSSM system (Fig 4D). On the negative dataset, the predicted false positive rate (FPR) in AAC, Hybrid was 0.005, and in PSSM 0.010.

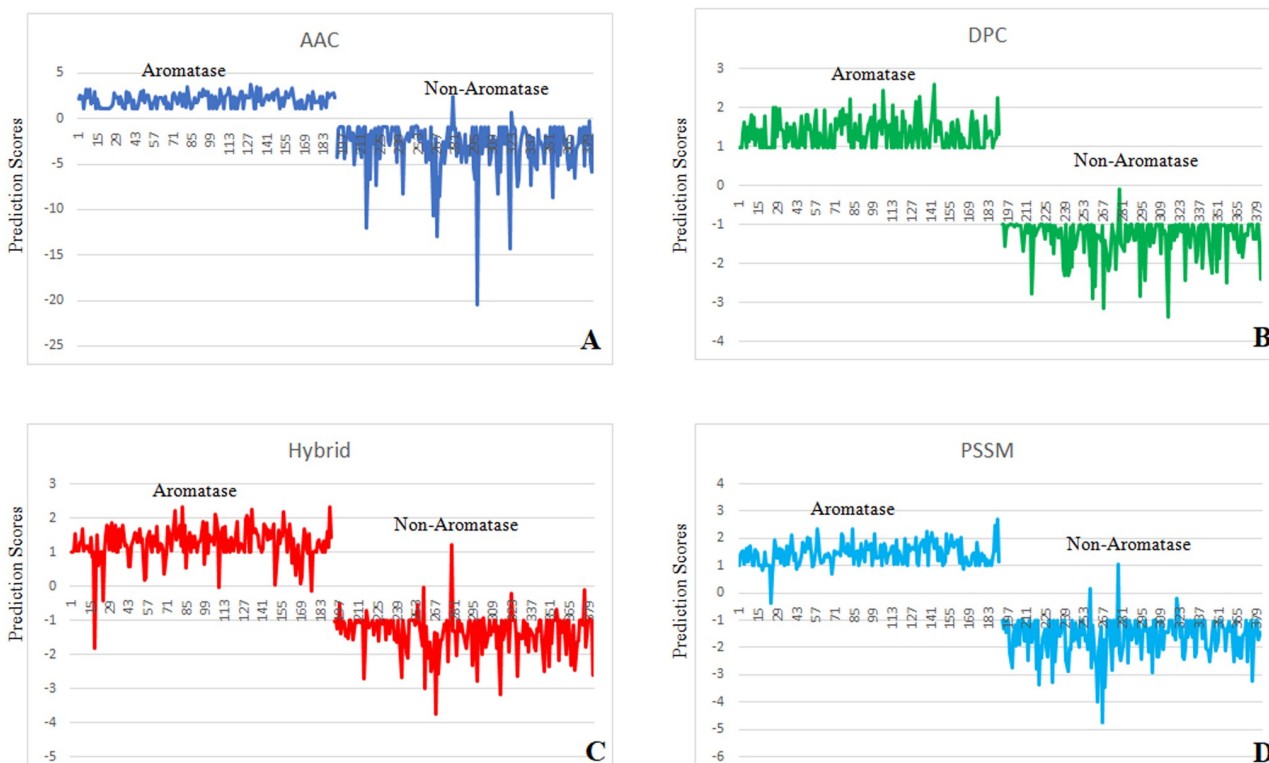

**Fig 4. Prediction scores graphs: Prediction performance of the developed models on aromatase and non-aromatase proteins.** A) Amino acid composition based approach (AAC), B) Dipeptide composition based approach (DPC), C) Hybrid profile based approach (AAC+DPC) and D) PSSM profile based approach.

## BLAST data analysis

According to the results of the BLAST dataset, the developed methods are performing well in all approaches in identifying aromatase. We have randomly picked five sequences from our dataset (CP19A_HUMAN, CP2F1_HUMAN, CP4Z1_HUMAN, GCM1_HUMAN, and CP2A7_HUMAN) and BLAST was performed against non-redundancy dataset and collected 500 sequences (100 each from one sequence). Overall, the proposed method using the BLAST dataset was able to accurately identify 97.4% of the sequences in all approaches. All models correctly predicted the respective individual performances of AAC, DPC, Hybrid and PSSM at 99.2%, 93.8%, 96.6% and 100% (Table 3). Thus, the PSSM approach completely identifies the BLAST sequences (Fig 5). This result shows that our method outperforms the BLAST search in identifying the aromatase related proteins.

**Table 3. The prediction performance of all models on the BLAST-Search data.**

|  | Total BLAST sequences | Positive Prediction | Negative Prediction | Positive Prediction Percentage |
|---|---|---|---|---|
| AAC | 500 | 496 | 4 | 99.2% |
| DPC | 500 | 469 | 31 | 93.8% |
| Hybrid (AAC+DPC) | 500 | 483 | 17 | 96.6% |
| PSSM | 500 | 500 | 0 | 100% |

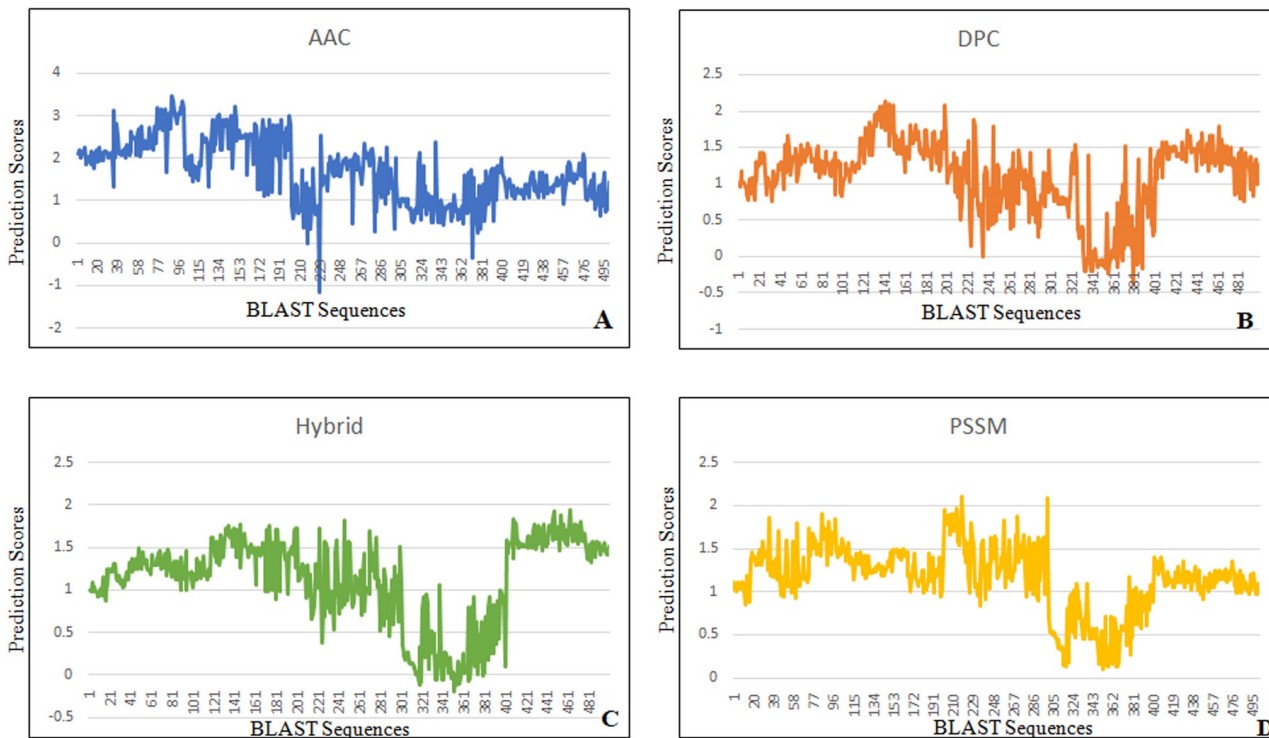

**Fig 5. BLAST-Search Data analysis: Prediction performance of all models on the BLAST-Search data, A) AAC, B) DPC, C) Hybrid and D) PSSM.**

## Discussion

Computational biology has helped understand proteins from a new perspective, as algorithms can predict protein-protein interactions [45, 46] and identify novel drug targets in various pathologies [47, 48]. Algorithms performing systematic study of cancer and protein databases [49, 50] have enhanced the accuracy of cancer patients' survival predictions [51–54], provide understanding of drug-induced side-effects [55] and allow identification of novel biomarkers [56]. To our knowledge, there are no algorithms for structural and functional characterization of aromatase or its polymorphisms. As aromatase is a critical target in breast cancer patients [57, 58], we established a reliable approach for detecting novel aromatase-related proteins, which will aid in developing novel AIs with improved efficacy.

Aromatase belong to the cytochrome P450 family, which are heme-containing mono-oxygenases and highly flexible enzymes that allow easy substrate access and binding, and product release [59]. Unlike most P450s, which are not highly substrate selective, androgenic specificity of aromatase sets it apart. Aromatase structure remained unknown for decades and this hindered explanation of its biochemical mechanism. Several laboratories purified aromatase from human placenta [60, 61] and recombinant expression systems [62, 63], however attempts to crystallize aromatase remained unsuccessful. So far, only one crystal structure of the only natural mammalian, full-length P450 human placental aromatase is known [64]. Thus, finding aromatase-related proteins using in-vivo and in-vitro methods is difficult and thus low-cost computational methods like SVM can be a reliable approach to identify novel aromatase-related proteins.

Aromatase is the only vertebrate enzyme which catalyzes aromatization of androgens into estrogens [64, 65]. It is a monomeric integral membrane protein in endoplasmic reticulum [66, 67] and has a heme group with 503 amino acids. Aromatase has twelve α-helices and ten β-strands [64, 68] and its active site is a distal cavity of heme-binding pocket with heme iron being the reaction center [68]. Aromatase in peripheral adipose tissues leads to estrogen biosynthesis in postmenopausal women, thus inducing breast tumors [69]. A small amount of estrogen can stimulate breast tumor formation and aromatase protein is seen in epithelial as well as stromal breast cancer cells [70]. AIs are currently being used to treat breast cancer patients, however resistance and toxicity of AIs induces the need for discovering novel AIs [71].

Survival analysis in various types of cancer patients using KM plotter showed that aromatase higher mRNA expression led to poorer overall survival (OS) in head-neck squamous cell carcinoma (Fig 1A), kidney renal clear cell carcinoma (Fig 1B), kidney renal papillary cell carcinoma (Fig 1C), liver hepatocellular carcinoma (Fig 1E) and stomach adenocarcinoma (Fig 1F) patients. Human fetal liver, kidney and intestine expresses significant level of aromatase [72], but the hepatic aromatase expression becomes untraceable in post-natal life [73]. Estrogens have shown to promote not only the development and progression of breast cancer, but also endometrial, prostrate and colorectal cancer by increasing the mitotic activity [74, 75]. The current survival analysis suggests a key role of aromatase as a tumor-promoter, even in extragonadal tissues including head-neck, kidney, liver and stomach [76]. These results signify the demand for a method to identify aromatase-related proteins for various types of endocrine-responsive tumors.

SVM is used in a variety of studies in the field of basic science and medicine, including clinical data analysis, laboratory testing for detection of disease and clinical trials of medicines [77–79]. In this study, we developed a very reliable method for predicting aromatase-related proteins, based on a variety of protein patterns such as AAC, DPC and Hybrid approaches. The overall prediction accuracy for aromatase-related proteins was 87.42%, 84.05%, 85.12% and 92.02% for AAC, DPC, hybrid and PSSM, respectively. The results of the BLAST search data analysis and prediction score graph analysis demonstrate that the established method is effective in identifying the aromatase-related proteins. We expect that our developed method will find undiscovered aromatase-related proteins, which will aid researchers in cancer predictive studies and precision medicine. As it is a first webserver to detect aromatase-related proteins, we cannot compare the performance of our method with any other methods.

## Conclusion

So far, there is no web-server/algorithm to predict or detect aromatase-related proteins. Thus, we developed a highly accurate method for identifying aromatase-related proteins using SVM with various amino acid approaches (Fig 6). The method was developed with the fivefold cross validation techniques with the approaches of amino acid composition (AAC), dipeptide composition (DPC), hybrid (AAC+DPC) and position specific score matrix (PSSM). We have tested the known and unknown data with our developed models and as a result all models detect aromatase-related proteins accurately. In future studies, we would like to work on the aromatase inhibitors with molecular docking, and we are also interested in using a deep learning technique [80–82]. We believe that this study will facilitate researchers in finding new or undiscovered aromatase-related proteins.

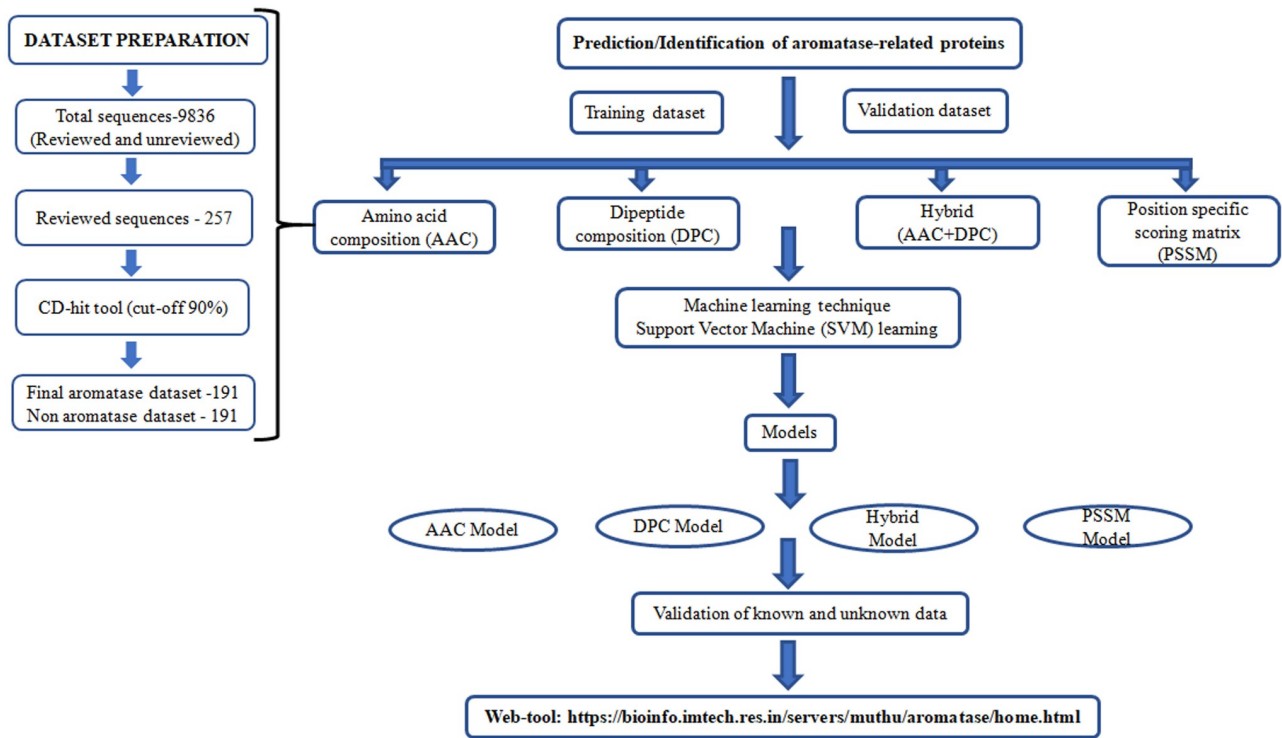

**Fig 6. Flow chart for developing SVM method to predict aromatase-related proteins.**

## Supporting information

**S1 File.**
(DOC)

## Acknowledgments

We are sincerely thankful to the Directors of CSIR-IMTECH and PGIMER (Chandigarh) for their support. A copy of the manuscript has been submitted to PTM, CSIR IMTECH, dated on 25.07.2022.

## Author Contributions

**Conceptualization:** Muthu Krishnan Selvaraj, Jasmeet Kaur.

**Data curation:** Muthu Krishnan Selvaraj, Jasmeet Kaur.

**Formal analysis:** Muthu Krishnan Selvaraj, Jasmeet Kaur.

**Investigation:** Muthu Krishnan Selvaraj, Jasmeet Kaur.

**Methodology:** Muthu Krishnan Selvaraj, Jasmeet Kaur.

**Project administration:** Muthu Krishnan Selvaraj, Jasmeet Kaur.

**Resources:** Muthu Krishnan Selvaraj, Jasmeet Kaur.

**Software:** Muthu Krishnan Selvaraj, Jasmeet Kaur.

**Supervision:** Muthu Krishnan Selvaraj, Jasmeet Kaur.

**Validation:** Muthu Krishnan Selvaraj, Jasmeet Kaur.

**Visualization:** Muthu Krishnan Selvaraj, Jasmeet Kaur.

**Writing – original draft:** Muthu Krishnan Selvaraj, Jasmeet Kaur.

**Writing – review & editing:** Muthu Krishnan Selvaraj, Jasmeet Kaur.

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
