## [Decision Letter · Decision Letter 0]

31 Oct 2022

PONE-D-22-24198Computational method for aromatase-related proteins using machine learning approachPLOS ONE

Dear Dr. Kaur,

Thank you for submitting your manuscript to PLOS ONE. After careful consideration, we feel that it has merit but does not fully meet PLOS ONE’s publication criteria as it currently stands. Therefore, we invite you to submit a revised version of the manuscript that addresses the points raised during the review process.

We look forward to receiving your revised manuscript.

Kind regards,

Avaniyapuram Kannan Murugan, M.Phil., Ph.D.

Academic Editor

PLOS ONE

Journal Requirements:

2. Please note that PLOS ONE has specific guidelines on code sharing for submissions in which author-generated code underpins the findings in the manuscript. In these cases, all author-generated code must be made available without restrictions upon publication of the work. Please review our guidelines at https://journals.plos.org/plosone/s/materials-and-software-sharing#loc-sharing-code and ensure that your code is shared in a way that follows best practice and facilitates reproducibility and reuse. New software must comply with the Open Source Definition.

Additional Editor Comments:

Reviewers positively comment on the manuscript. However, raise many major critiques on the manuscript. Kindly address them carefully giving additional input.

Reviewers' comments:

Reviewer's Responses to Questions

**Comments to the Author**

1. Is the manuscript technically sound, and do the data support the conclusions?

Reviewer #1: Yes

Reviewer #2: Partly

Reviewer #3: Yes

Reviewer #4: No

2. Has the statistical analysis been performed appropriately and rigorously? 

Reviewer #1: I Don't Know

Reviewer #2: Yes

Reviewer #3: Yes

Reviewer #4: Yes

3. Have the authors made all data underlying the findings in their manuscript fully available?

Reviewer #1: Yes

Reviewer #2: Yes

Reviewer #3: Yes

Reviewer #4: No

4. Is the manuscript presented in an intelligible fashion and written in standard English?

Reviewer #1: No

Reviewer #2: No

Reviewer #3: Yes

Reviewer #4: No

5. Review Comments to the Author

Reviewer #1: The manuscript sounds interesting, and the authors have developed method to predict the aromatase-related proteins. The manuscript is well structured and the authors have elucidated their method distinctly. However, There are some corrections could be made to make it more precise and clear;

1- The abstract should be more organized and includes briefly the method, result, and conclusion.

2- The last paragraph of the introduction should be moved to the method section.

3- There are some out dated references, I recommend the authors to use most recent ones.

Reviewer #2: Dear authors,

Your manuscript is highly interesting, and it caught my interest as I wrote almost same journal article after a regress work on SARS CoV-2. However, your manuscript is having a huge flaw, the Abstract, methods, and conclusion are totally not in the article fashion, improvise it properly as you have done wonderful work.

Reviewer #3: Dear Authors,

Thank you for submitting your work to Plos One journal. Some improvements need to be considered to make your work looks better.

Please consider the following comments and suggestions:

1. Abstract:

a. I suggest moving the link to the related section (dataset).

b. Please recheck the link and make sure it is correct and working.

2. Introduction:

Please follow the traditional way of writing this part. For instance, at the end of this part you should mention the structure of your article.

3. Methods:

- Datasets for SVM:

you mentioned removing sequences labeled "fragments,", "isoforms",.. etc. Please elaborate.

- Generation of survival curves:

please explain more about Kaplan-Meier tool.

4. Results:

- In table 1:

The p-value numbers varies a lot! for example the p-value for Esophageal Adenocarcinoma on RFS was 0.93 while o-value for Head-neck squamous cell carcinoma was 0.048. Please explain the reason.

5. Conclusion:

Based on your findings, do you suggest any future work?

- Please check the equations and formulas because it appears that there's something wrong with the fraction bars!

Reviewer #4: The present study aims to develop a method to recognize aromatase protein using different amino acid composition parameters and evolutionary profile using SVM technique. Also, the authors state that they have developed a webserver for the same.

However, from the introduction given in the manuscript it is difficult to understand that why authors need to predict Aromatase protein. They state that Aromatase is a target for designing inhibitors in humans for treating Cancer and other diseases. But, then the sequence of this protein is already known. So what is the purpose of this study.

Also, if I understand Aromatase is one single protein and if we start developing methods for a single proteins then we can develop many thousand methods. But what is the utility?

Minor comments

1. In “Method” section, number of sequences initially taken from Uniprot/Swiss-prot as well as number of sequences removed to get a final dataset of 191 sequences is not mentioned.

2. Have the authors compared the similarity of negative set sequences with positive dataset?

3. Amino acid profiles of aromatase and non-aromatase was found to be similar by authors (Fig 2B). Then how it has contributed in model devlopment.

4. The web link given for the server could not accessed.

6. PLOS authors have the option to publish the peer review history of their article (what does this mean?). If published, this will include your full peer review and any attached files.

Reviewer #1: **Yes: **Alhassan Ali Ahmed

Reviewer #2: **Yes: **Shaban Ahmad

Reviewer #3: **Yes: **Abdullah Almuhaimeed

Reviewer #4: No

---

## [Author Response · Author response to Decision Letter 0]

10 Jan 2023

Dear Editor,

Thank you for providing us an opportunity to submit a revision of our manuscript. Below is a point-by-point response to all the reviewers. All the changes in the revised manuscript are highlighted in yellow. 

Reviewer #1: 

The manuscript sounds interesting, and the authors have developed method to predict the aromatase-related proteins. The manuscript is well structured and the authors have elucidated their method distinctly. However, there are some corrections could be made to make it more precise and clear;

1- The abstract should be more organized and includes briefly the method, result, and conclusion.

Yes, we have modified the abstract as per your guidance (highlighted in yellow). 

2- The last paragraph of the introduction should be moved to the method section.

Yes, some part of the paragraph is now moved to the Method section on page 4 as para 1 (highlighted in yellow). 

3- There are some out dated references, I recommend the authors to use most recent ones.

Yes, now we have added the recent references (highlighted in yellow).

Reviewer #2: 

Dear authors,

Your manuscript is highly interesting, and it caught my interest as I wrote almost same journal article after a regress work on SARS CoV-2. However, your manuscript is having a huge flaw, the Abstract, methods, and conclusion are totally not in the article fashion, improvise it properly as you have done wonderful work.

Yes, we have modified the abstract, methods and conclusion as per your guidance (highlighted in yellow). 

Reviewer #3: 

Dear Authors,

Thank you for submitting your work to Plos One journal. Some improvements need to be considered to make your work looks better.

Please consider the following comments and suggestions:

1. Abstract:

a. I suggest moving the link to the related section (dataset).

Yes, we have moved the link to webserver in dataset section in the methods on page 5 (highlighted in yellow). 

b. Please recheck the link and make sure it is correct and working.

Due to some security reason the provided link was not working, now we have secured our site under (https). Now it can be accessed. 

2. Introduction:

Please follow the traditional way of writing this part. For instance, at the end of this part you should mention the structure of your article.

Yes, we have modified the end of Introduction as per your guidance (highlighted in yellow). 

3.Methods:

- Datasets for SVM: you mentioned removing sequences labeled "fragments,", "isoforms",.. etc. Please elaborate.

Yes, it is now explained in the Dataset for SVM section on page 5 (highlighted in yellow). 

- Generation of survival curves: please explain more about Kaplan-Meier tool.

Yes, the explanation is now added to the section on page 4 (highlighted in yellow). 

4. Results:

- In table 1: The p-value numbers varies a lot! for example the p-value for Esophageal Adenocarcinoma on RFS was 0.93 while o-value for Head-neck squamous cell carcinoma was 0.048. Please explain the reason.

The p-value is calculated by the KM plotter website for each cancer type and hence it is significant for some types of cancers and not significant for others.

5. Conclusion:

Based on your findings, do you suggest any future work?

Yes, we are interested to work on molecular docking of aromatase inhibitors.

- Please check the equations and formulas because it appears that there's something wrong with the fraction bars!

Yes, we have corrected the equations and formulas.

Reviewer #4: The present study aims to develop a method to recognize aromatase protein using different amino acid composition parameters and evolutionary profile using SVM technique. Also, the authors state that they have developed a webserver for the same.

However, from the introduction given in the manuscript it is difficult to understand that why authors need to predict Aromatase protein.

There are many unknown functional proteins, which are available and need to be identified. Thus, there is a need of computational methods to identify these proteins and their functions. So, we have developed a method to identify these aromatase-related proteins. 

They state that Aromatase is a target for designing inhibitors in humans for treating Cancer and other diseases. But, then the sequence of this protein is already known. So what is the purpose of this study. Also, if I understand Aromatase is one single protein and if we start developing methods for single proteins then we can develop many thousand methods. But what is the utility?

It is not like that; we have used aromatase as one of the major enzymes in the cancer-related study. So, we developed a computational method which will be useful for the cancer researchers. Actually, it is one of the major tasks in bioinformatics to predict the protein functions which can assist with a variety of biological issues, such as understanding disease mechanisms or identifying novel drug targets.

Minor comments

1. In “Method” section, number of sequences initially taken from Uniprot/Swiss-prot as well as number of sequences removed to get a final dataset of 191 sequences is not mentioned.

Yes, when we used the keyword “aromatase”, we found 9836 protein sequences which included 257 reviewed sequences. We only used reviewed sequences and the final dataset contained 191 sequences out of 257. This is added on page 5 under “Datasets for SVM” (highlighted in yellow).

2. Have the authors compared the similarity of negative set sequences with positive dataset?

We have not done the similarity of negative set sequences with positive dataset. The negative sequences are totally different from positive set sequences which were selected using keyword in the uniprot databases. 

3. Amino acid profiles of aromatase and non-aromatase was found to be similar by authors (Fig 2B). Then how it has contributed in model development

Sorry, some of the residues patterns are similar, but not all. These differences can identify the aromatase from negative sequence by the developed models.

4. The web link given for the server could not accessed.

Due to some security reason the provided link was not working, now we have secured our site under (https). Now it can be accessed.

---

## [Decision Letter · Decision Letter 1]

10 Feb 2023

PONE-D-22-24198R1Computational method for aromatase-related proteins using machine learning approachPLOS ONE

Dear Dr. Kaur,

Thank you for submitting your manuscript to PLOS ONE. After careful consideration, we feel that it has merit but does not fully meet PLOS ONE’s publication criteria as it currently stands. Therefore, we invite you to submit a revised version of the manuscript that addresses the points raised during the review process.

We look forward to receiving your revised manuscript.

Kind regards,

Avaniyapuram Kannan Murugan, M.Phil., Ph.D.

Academic Editor

PLOS ONE

Reviewers' comments:

Reviewer's Responses to Questions

**Comments to the Author**

1. If the authors have adequately addressed your comments raised in a previous round of review and you feel that this manuscript is now acceptable for publication, you may indicate that here to bypass the “Comments to the Author” section, enter your conflict of interest statement in the “Confidential to Editor” section, and submit your "Accept" recommendation.

Reviewer #1: All comments have been addressed

Reviewer #2: (No Response)

Reviewer #3: All comments have been addressed

2. Is the manuscript technically sound, and do the data support the conclusions?

Reviewer #1: Yes

Reviewer #2: Yes

Reviewer #3: Yes

3. Has the statistical analysis been performed appropriately and rigorously? 

Reviewer #1: I Don't Know

Reviewer #2: Yes

Reviewer #3: Yes

4. Have the authors made all data underlying the findings in their manuscript fully available?

Reviewer #1: Yes

Reviewer #2: No

Reviewer #3: Yes

5. Is the manuscript presented in an intelligible fashion and written in standard English?

Reviewer #1: Yes

Reviewer #2: Yes

Reviewer #3: Yes

6. Review Comments to the Author

Reviewer #1: (No Response)

Reviewer #2: Dear Authors,

I see the major revisions that you have made so far, and I provided only basic suggestions previously and now providing the extensive suggestions. Kindly make the possible changes to make the manuscript good for the reader and future researchers.

1. Make a graphical abstract (Flow chart) for the complete study how you conceptualised to what you obtained followed by each method in depth. (https://www.ncbi.nlm.nih.gov/core/lw/2.0/html/tileshop_pmc/tileshop_pmc_inline.html?title=Click%20on%20image%20to%20zoom&p=PMC3&id=6386390_pcbi.1006730.g001.jpg)

2. Kindly make the source code available through GitHub with proper documentations.

3. The server link is not working, make sure your institute keep it live.

4. You have used different SVM approach-based models with amino acid, dipeptide composition, hybrid and evolutionary profiles to predict. This is good; however, the tool must work with it in background and in the front the user only supposed to see the results and if user wants there supposed to be a button to view the documentation how and why the SVM has classified them to be aromatase.

5. In methods- Provide the dataset in supplementary files and keep the search date to not make it irrelevant for future researchers.

6. The Introduction and discussion is not sufficiently written, it needed to be validated with any other expert, authors can read some more articles to understand how to write introduction as well as discussion.

I wish all the best to authors for the revision and hoping they will make it asap.

Reviewer #3: Dear Authors,

Thank you for re-submitting your work to PLOS ONE journal. All my comments and suggestions have been covered successfully.

Thank you

7. PLOS authors have the option to publish the peer review history of their article (what does this mean?). If published, this will include your full peer review and any attached files.

Reviewer #1: No

Reviewer #2: **Yes: **

Reviewer #3: **Yes: **

---

## [Author Response · Author response to Decision Letter 1]

28 Feb 2023

Dear Editor,

Thank you for providing us an opportunity to submit a revision of our manuscript. Below is a point-by-point response to the reviewer. All the changes in the revised manuscript are highlighted in yellow. 

Reviewer 2: 

1. Make a graphical abstract (Flow chart) for the complete study how you conceptualized to what you obtained followed by each method in depth. https://www.ncbi.nlm.nih.gov/core/lw/2.0/html/tileshop_pmc/tileshop_pmc_inline.html?title=Click%20on%20image%20to%20zoom&p=PMC3&id=6386390_pcbi.1006730.g001.jpg)

Yes, we have prepared a graphical abstract as per your guidance, which is now Figure No. 6. 

2. Kindly make the source code available through GitHub with proper documentations.

It is really good to keep our web-tool in the GitHub and we will try it in our future studies. At present, our institute (CSIR-IMTECH) has provided all the facility to host our web-tool and it is also being a part of an open source portal. 

3. The server link is not working, make sure your institute keep it live.

Yes, our server is on live at https://bioinfo.imtech.res.in/servers/muthu/aromatase/submit.html

4. You have used different SVM approach-based models with amino acid, dipeptide composition, hybrid and evolutionary profiles to predict. This is good; however, the tool must work with it in background and in the front the user only supposed to see the results and if user wants there supposed to be a button to view the documentation how and why the SVM has classified them to be aromatase.

Yes we agree, but it is not possible to show all the background running programs. Actually, it has a set of programs wrote in perl script. All the programs were kept in cgi-bin folder, which run only in the background and it is not for public viewing. The output results will be displayed in a new web page. 

From documentation point of view, it is a good idea that the user can view the complete results. We will try it in our future studies.

5. In methods - Provide the dataset in supplementary files and keep the search date to not make it irrelevant for future researchers.

Yes, we have provided all the dataset as a supplementary file-1. The retrieval date is now mentioned in the dataset section (highlighted in yellow). 

6. The Introduction and discussion is not sufficiently written, it needed to be validated with any other expert, authors can read some more articles to understand how to write introduction as well as discussion.

Yes, we have modified the Introduction and discussion as per your guidance (highlighted in yellow).

---

## [Decision Letter · Decision Letter 2]

13 Mar 2023

Computational method for aromatase-related proteins using machine learning approach

PONE-D-22-24198R2

Dear Dr. Kaur,

We’re pleased to inform you that your manuscript has been judged scientifically suitable for publication and will be formally accepted for publication once it meets all outstanding technical requirements.

Kind regards,

Avaniyapuram Kannan Murugan, M.Phil., Ph.D.

Academic Editor

PLOS ONE

Additional Editor Comments (optional):

Reviewers' comments:

Reviewer's Responses to Questions

**Comments to the Author**

1. If the authors have adequately addressed your comments raised in a previous round of review and you feel that this manuscript is now acceptable for publication, you may indicate that here to bypass the “Comments to the Author” section, enter your conflict of interest statement in the “Confidential to Editor” section, and submit your "Accept" recommendation.

Reviewer #2: All comments have been addressed

Reviewer #3: All comments have been addressed

2. Is the manuscript technically sound, and do the data support the conclusions?

Reviewer #2: Yes

Reviewer #3: Yes

3. Has the statistical analysis been performed appropriately and rigorously? 

Reviewer #2: N/A

Reviewer #3: Yes

4. Have the authors made all data underlying the findings in their manuscript fully available?

Reviewer #2: Yes

Reviewer #3: Yes

5. Is the manuscript presented in an intelligible fashion and written in standard English?

Reviewer #2: Yes

Reviewer #3: Yes

6. Review Comments to the Author

Reviewer #2: Dear authors,

I went through the revisions, and found it suitable. I mark it for acceptance, rest it depends on the Editor's decision as the final.

Reviewer #3: Dear Authors,

Thank you for re-submitting your work to PLOS ONE journal. All comments and suggestions have been covered successfully.

Thank you

7. PLOS authors have the option to publish the peer review history of their article (what does this mean?). If published, this will include your full peer review and any attached files.

Reviewer #2: **Yes: **

Reviewer #3: **Yes: **

---

## [Editor Report · Acceptance letter]

17 Mar 2023

PONE-D-22-24198R2 

Computational method for aromatase-related proteins using machine learning approach 

Dear Dr. Kaur:

I'm pleased to inform you that your manuscript has been deemed suitable for publication in PLOS ONE. Congratulations! Your manuscript is now with our production department. 

Kind regards, 

on behalf of

Dr. Avaniyapuram Kannan Murugan 

Academic Editor

PLOS ONE